# Optimizing wheelchair basketball lineups: A statistical approach to coaching strategies

**Valentina Cavedon**[1]*, **Paola Zuccolotto**[2], **Marco Sandri**[2], **Maricay Manisera**[2], **Marco Bernardi**[3,4], **Ilaria Peluso**[5], **Chiara Milanese**[1]

**1** Department of Neurosciences, Biomedicine and Movement Sciences, University of Verona, Verona, Italy, **2** BODaI-Lab, University of Brescia, Brescia, Italy, **3** Department of Physiology and Pharmacology "V. Erspamer", Sapienza University of Rome, Rome, Italy, **4** Italian Wheelchair Basketball Federation, Rome, Italy, **5** Research Centre for Food and Nutrition, Council for Agricultural Research and Economics, CREA-AN, Rome, Italy

\* valentina.cavedon@univr.it

**Data Availability Statement:** All relevant data are within the manuscript and its Supporting Information files.

## Abstract

This study was designed to support the tactical decisions of wheelchair basketball (WB) coaches in identifying the best players to form winning lineups. Data related to a complete regular season of a top-level WB Championship were examined. By analyzing game-related statistics from the first round, two clusters were identified that accounted for approximately 35% of the total variance. Cluster 1 was composed of low-performing athletes, while Cluster 2 was composed of high-performing athletes. Based on data related to the second round of the Championship, we conducted a two-fold evaluation of the clusters identified in the first round with the team's net performance as the outcome variable. The results showed that teams where players belonging to Cluster 2 had played more time during the second round of the championship were also those with the better team performance (R-squared = 0.48, $p$ = 0.035), while increasing the playing time for players from Classes III and IV does not necessarily improve team performance ($r2$ = -0.14, $p$ = 0.59). These results of the present study suggest that a collaborative approach between coaches and data scientists would significantly advance this Paralympic sport.

## Introduction

Wheelchair Basketball (WB) is one of the most popular Paralympic sports, and it is played on a competitive level in over 80 countries around the world [1]. As highlighted by Askew and colleagues [2], high-performance Paralympic sports provide rich environments for innovation also because performance strategies and coaching methods developed in an able-bodied context cannot simply be transplanted into the Paralympic setting [3]. Since WB has reached an increased level of professionalism, the primary concerns of the coaches and the technical staff are how to optimize the factors contributing to the success of game performance and how to select players [4, 5].

To the best of our knowledge, support tools providing evidence to help coaches and technical staff in selecting the players to be deployed on the field during a WB game are still missing.

**Funding:** Research Project PRIN 2022, granted by European Union – Next Generation EU, "Statistical Models and AlgoRiThms in sports (SMARTsports). Applications in professional and amateur contexts, with able-bodied and disabled athletes", project nr. 2022R74PLE, CUP: D53D23005950006.

**Competing interests:** The authors have declared that no competing interests exist.

Accordingly, today, the coach's decision on who to put on the field during a game is primarily driven by a mix of personal feelings based on the player's technical skills and their chemistry with their teammates. This decision making is of particular interest in a team sport like WB, where players are assigned a functional point score (herein after point) from point 1.0 (i.e., players with minimal functional potential, like players with a complete spinal cord injury at the thoracic level) to point 4.5 (i.e., players with maximal functional potential, like players with unilateral below-knee amputation). These points are on an ordinal scale with 0.5-point increments, and coaches must consider that during the game, the sum of the points of the five players on the court cannot exceed 14.0 [6]. So, this constraint is one of the peculiarities of WB that makes the coach's decisions a challenge.

In recent years, with the rapid development of computer technology, the data scale of sports, especially in running basketball, has increased sharply, promoting the advent of the era of sports big data [7]. Data science applied to sports is rapidly gaining interest [8–11], and today, more and more coaches, players, scouts, and sports managers recognize its value as a reliable support in making decisions during the games.

In this field, data mining algorithms define methods and rules to split major data into groups, which are not defined *a priori* and are supposed to somehow reflect the structure of the entities that the data represent [12]. The unmanipulated classification of individual cases into groups whose profile design spontaneously emerges from data is known as cluster analysis [13]. For example, cluster analysis has been employed in running basketball to cluster players, teams, matches and game fractions [14–16]. In analogy to the above-reported literature on running basketball, it is reasonable to assume that applying cluster analysis in WB game performance data may help coaches in their tactical choices and in the management of a championship.

While the use of data science for various purposes is well established in running basketball [17], a smaller number of contributions can be found in the literature regarding WB [18, 19]. This is mainly due to the unavailability of the large and complete datasets that are instead often guaranteed in running basketball. Today, game performance in WB can be represented by season statistics regarding winning records, average points from field-goals and free throws, rebounds, assists, and steals per match [4, 20]. Research describing game performance in WB mainly focused on players' classification points and playing positions, gender differences, game types (i.e., balanced vs unbalanced teams) and discriminating factors between successful and unsuccessful teams [1, 4, 21–25].

In line with the literature on running basketball [26–30], collecting large quantities of data which describe game performance in WB and the appropriate analysis of this data using statistical methods can provide helpful information to support coaches in their decision making [17]. As a first step toward this aim, this study was designed to provide a practical tool for WB coaches based on statistical techniques to support their tactical decisions in identifying the most influential players to form winning lineups while satisfying the 14-point constraint. We will show that, generally, the players with a high point are not necessarily those with the best overall performance, as measured by a set of game statistics. We go on to develop a method that enables recognising the best-performing players regardless of their level of functional ability. This can be of utmost use, especially for identifying players who can be more valuable to the team while still adhering to the 14-point constraint.

## Materials and methods

### Study design

In line with the aims of our study, we applied a non-participative approach. This study considered the regular season of the top Italian Wheelchair Basketball Championship ("Serie A"). It

is managed by the Federazione Italiana Pallacanestro in Carrozzina (FIPIC, the Italian Wheelchair Basketball Federation), which set out the technical and medical regulations governing the championship under the surveillance of the Italian Paralympic Committee, the International Paralympic Committee, and the International Wheelchair Basketball Federation. Like other countries, the regular season of the top-level Italian Wheelchair Basketball Championship employs a double round-robin format (i.e., a first round and a second round) [31]. In this system, each team faces every other team twice: once at their home venue and once as visitors at the opposing team's venue. The regular season analyzed consisted of 16 championship days across a six month period (from November to April), during each of which four matches were held, for a total of 64 games played by eight teams.

**Sample.** Data related to the 2018–2019 regular season of the top Italian Wheelchair Basketball Championship were extracted from the official score sheets provided by the FIPIC. During the observed competitive season, 101 athletes from eight teams participated in the premier Italian Wheelchair Basketball Championship. The athletes included males (n = 94) and females (n = 7), with an average age of 32.7 ± 9.3 years. The distribution of participants across the assigned point range was as follows: 1.0 points (n = 19), 1.5 points (n = 10), 2.0 points (n = 8), 2.5 points (n = 17), 3.0 points (n = 9), 3.5 points (n = 7), 4.0 points (n = 22) and 4.5 points (n = 9). For this study, players were distributed across four functional classes as follows: Class I (1.0–1.5 points, n = 29), Class II (2.0–2.5 points, n = 25), Class III (3.0–3.5 points, n = 16), and Class IV (4.0–4.5 points, n = 31). Inclusion criteria was playing at least two championship matches during the season (i.e., at least one match during the first round and at least one match during the second round). "Playing" was defined as being on the court at least once during a game.

**Procedures.** From the score sheet of each match, we considered the following game-related statistics per athlete: the number of free-throw points made (FTM), the number of two-point field-goals made (P2M), the number of three-point field-goals made (P3M), the total points made per match (PTS = FTM + 2*P2M + 3*P3M), the number of steals (ST), the number of rebounds (REB) and the number of assists (AS) (Table 1). These variables were normalized by the time (expressed in minutes) each player spent on the court during each match. General characteristics of athletes (i.e., age, sex, and assigned Points [1.0–4.5]) were obtained from the database available on the FIPIC website [32].

## Statistical analysis

Data for continuous variables were summarized using the median and interquartile range (IQR), while absolute and relative frequencies were used for categorical variables. The Mann-Whitney test was employed for two-group comparisons of continuous variables, the Kruskal-Wallis test for comparisons among the four groups (i.e., Class I, Class II, Class III and Class IV), and Fisher's exact test was used to compare distributions of categorical variables.

**Table 1. Game-related statistics.**

| Variable | Abbreviation |
|---|---|
| Free-throw points made | FTM |
| Two-point field-goals made | P2M |
| Three-point field-goals made | P3M |
| Total points made per match | PTS |
| Steals | ST |
| Rebounds | REB |
| Assist | AS |

The analysis performed in our study consisted of three key steps: Clustering, Characterization, and Validation. Cluster analysis is a statistical method that groups a set of objects in such a way that objects in the same group (called a cluster) are more similar to each other than to those in other groups. In other words, this technique can identify groups with minimal within-cluster variability and significant between-cluster variability [33]. Clustering is an inherently multivariate task, thus it is beyond the capabilities of the human brain. In our work, we applied the k-means clustering method described by Zuccolotto and Manisera [17] to seven game-related statistics (FTM, P2M, P3M, PTS, ST, REB, and AS) from the first-round dataset, utilizing radial plots of the average profile to visualize and compare the patterns of mean values of the game-related statistics across the identified clusters [17].

In the Characterization step of our analysis, we utilized data from the second round of the championship to conduct a descriptive analysis, which enhanced our understanding of the nature of the identified clusters. This step also served as an initial informal validation by assessing the out-of-sample characteristics of the clusters.

The final step, Validation, involved investigating the connections between teams' performance and the groupings identified by clustering, using data from the second round. We estimated two linear regression models, both using the team's net performance (i.e., the difference between the points scored by the team and those scored by the opposition) as the outcome variable. The first model (Model 1) used the minutes played by players from Cluster 2 as the predictor variable. For the second model (Model 2), the covariates were the minutes played by team players in three of the four functional classes (Class II, III, and IV). We excluded the variables representing minutes in Cluster 1 and in Class I from these models due to their perfect collinearity with the other covariates. Using these models, we could evaluate whether teams that allocate more playtime to players from a particular group have better performance outcomes. The procedures adopted are summarized in the diagram depicted in Fig 1.

The analyses were conducted using R version 4.3.2 (R Foundation for Statistical Computing, Vienna, Austria).

## Results

A total of 23 players were excluded from the analysis as they did not meet the inclusion criterion, which required playing at least two championship matches during the season (see Materials and Methods section).

### Identifying high performers in the first round

The distributions of the seven game-related statistics (which describe the athletes' contributions to the match) across the four functional classes are summarized in Table 2 for the first and second rounds. In the first round, functional classes III and IV exhibited significantly higher values of P2M, FTM, PTS, and REB compared to classes I and II, with $p$ ranging from 0.018 to less than 0.001. This pattern was also observed in the second round, with the additional inclusion of ST, all of which had $p$ of 0.003 or less.

By analyzing the seven game-related statistics (normalized by play time) from the first round of data, two clusters were identified that accounted for approximately 35% of the total variance. The distributions of these statistics across the identified clusters are summarized in Table 3. The profile plots of the two clusters are depicted in Fig 2. The left radial plot illustrates that all variable values for players within Cluster 1 are lower compared to those in Cluster 2, as depicted in the right-hand radial plot. Indeed, play time and all seven variables exhibited higher values in Cluster 2 identifying it as a group of high-performing athletes. All $p$ were significant, with values less than 0.002. No statistically significant differences were found in age

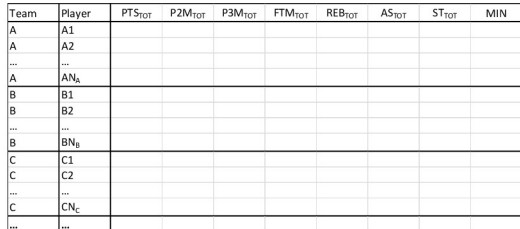

Collect players box scores data for the first
round of the Championship (all the teams)

Normalize by
minutes played

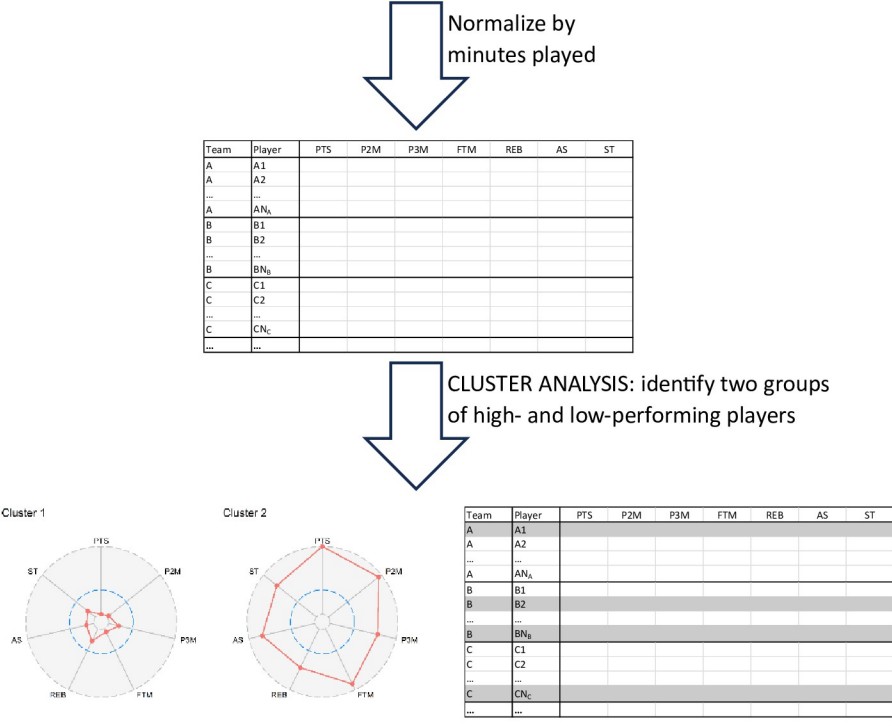

CLUSTER ANALYSIS: identify two groups
of high- and low-performing players

| | In the second round of the Championship | |
|---|---|---|
| **1** | Identify which of your team's players belong to the cluster of high-performing players. Should one player with low functional points belong to the cluster of high-performing players, this information is particularly relevant, as it allows to build high-performing lineups more likely to obey the 14-score constraint. |
| **2** | Identify which of your opponent team's players belong to the cluster of high-performing players. Use this information to define an optimal defense strategy |

**Fig 1. A schematic representation of the procedures adopted for the analysis conducted in this paper, illustrating the potential application of the proposed method in a real-world setting.**

**Table 2. Distribution of demographic and performance characteristics across four functional classes in the considered regular season.**

| Variable | Total (n = 78) | Class I (n = 21) | Class II (n = 17) | Class III (n = 12) | Class IV (n = 28) | p |
|---|---|---|---|---|---|---|
| Age (years) | 31 (27–38) | 31 (27–39) | 30 (27–36) | 35 (29–37) | 32 (26–37) | 0.8 |
| Sex (male) | 74 (95) | 19 (90) | 15 (88) | 12 (100) | 28 (100) | 0.2 |
| Play time (min) | 128 (39–186) | 108 (36–152) | 98 (40–200) | 161 (122–198) | 123 (52–170) | 0.4 |
| First round | | | | | | |
| PTS (n) | 29 (5–57) | 10 (4–28) | 18 (3–45) | 45 (19–62) | 35 (15–88) | 0.018 |
| P2M (n) | 12 (2–24) | 5 (2–13) | 9 (1–21) | 19 (9–24) | 16 (7–39) | 0.014 |
| P3M (n) | 0 (0–1) | 0 (0–0) | 0 (0–2) | 0 (0–5) | 0 (0–1) | 0.3 |
| FTM (n) | 2 (0–6) | 0 (0–1) | 1 (0–6) | 6 (1–9) | 4 (0–10) | 0.004 |
| REB (n) | 13 (3–28) | 5 (2–11) | 9 (2–21) | 28 (21–37) | 28 (9–43) | <0.001 |
| AS (n) | 6 (0–14) | 1 (0–9) | 10 (1–20) | 7 (4–15) | 6 (1–15) | 0.2 |
| ST (n) | 1 (0–3) | 0 (0–2) | 1 (0–1) | 2 (1–4) | 2 (0–5) | 0.051 |
| Second round | | | | | | |
| Play time (min) | 162 (67–237) | 116 (67–163) | 110 (67–208) | 240 (155–302) | 185 (41–237) | 0.1 |
| PTS (n) | 38 (6–81) | 13 (5–20) | 15 (6–52) | 63 (51–113) | 74 (11–107) | 0.003 |
| P2M (n) | 17 (3–34) | 6 (2–10) | 7 (3–19) | 28 (23–44) | 32 (5–47) | 0.001 |
| P3M (n) | 0 (0–1) | 0 (0–0) | 0 (0–1) | 0 (0–2) | 0 (0–1) | 0.2 |
| FTM (n) | 3 (0–7) | 0 (0–2) | 1 (0–7) | 6 (3–10) | 6 (1–16) | 0.002 |
| REB (n) | 16 (4–40) | 6 (3–10) | 8 (4–19) | 41 (32–52) | 37 (7.5–62) | <0.001 |
| AS (n) | 5 (1–20) | 3 (1–8) | 3 (0–18) | 13 (3–32) | 9 (2–27) | 0.08 |
| ST (n) | 1 (0–5) | 0 (0–1) | 1 (0–2) | 5 (2–8.5) | 4 (1–6) | 0.001 |

Data are summarized using the median and, in brackets, the interquartile range for continuous variables and absolute and percentage relative frequencies for categorical variables. $p$, p-value; min, minutes; PTS, total points made per match (PTS = FTM + 2*P2M + 3*P3M); P2M, number of two-point field-goals made; P3M, number of three-point field-goals made; FTM, number of free-throw points made; REB, number of rebounds; AS, number of assists; ST, number of steals.

and sex between the two clusters. Solutions with a higher number of clusters have also been explored. However, identifying more specific players' profiles did not result in a real improvement concerning the association of the clusters with the team's overall performance. The choice of two clusters, instead, allowed us to build a simple (easy to interpret) and effective tool with the aim of helping the coach in selecting the lineup.

The distribution of athletes across the four functional classes within the two clusters is illustrated in Fig 3. The two clusters show different distributions. Notably, Cluster 2, characterized by higher performance, is mainly composed of athletes from functional classes III and IV (82%), while 66% of the athletes in Cluster 1 belong to Classes I and II (P<0.001).

**Analysis of cluster characteristics in round two.** In the second step of the analysis, we investigated and compared the second-round attributes of the clusters identified in the first round. The latter part of Table 3 summarizes these characteristics. Interestingly, Cluster 2 still exhibited significantly higher values for all the seven game-related statistics.

**Second-round team performances based on functional classes and cluster membership.** In the last step of our analysis, we conducted a two-fold evaluation of the clusters identified in the initial round.

Firstly, we carried out a team-level validation of the clusters by deploying a linear regression model with each team's net performance as the outcome variable and the minutes played by players from Cluster 2 as the predictor variable. The analysis evidenced a statistically significant positive association between the two variables, with an adjusted $r^2$ of 0.48 and a $p$ of 0.026 (Table 4). This suggests that the more playing time given to players from Cluster 2, the better the team performs.

**Table 3. Distribution of demographic and performance characteristics across subgroups identified by cluster analysis applied to data of the first round.**

| Variable | Cluster 1 (n = 50) | Cluster 2 (n = 28) | *p* |
|---|---|---|---|
| Age (years) | 31 (27–39) | 31 (28–36) | 0.8 |
| Sex (male) | 46 (92) | 28 (100) | 0.3 |
| Class I (n) | 21 (42) | 0 (0) | <0.001 |
| Class II (n) | 12 (24) | 5 (18) | |
| Class III (n) | 7 (14) | 5 (18) | |
| Class IV (n) | 10 (20) | 18 (64) | |
| **First round** | | | |
| Play time (min) | 90 (32–157) | 155 (116–200) | 0.002 |
| PTS (n) | 12 (2–29) | 70 (45–100) | <0.001 |
| P2M (n) | 6 (1–13) | 28 (19–42) | <0.001 |
| P3M (n) | 0 (0–0) | 1 (0–2) | <0.001 |
| FTM (n) | 0 (0–3) | 8 (5–11) | <0.001 |
| REB (n) | 6 (1–15) | 29 (18–45) | <0.001 |
| AS (n) | 1 (0–7) | 14 (7–21) | <0.001 |
| ST (n) | 0 (0–1) | 3 (2–7) | <0.001 |
| **Second round** | | | |
| Play time (min) | 103 (49–193) | 207 (172–282) | <0.001 |
| PTS (n) | 13 (5–42) | 102 (71–124) | <0.001 |
| P2M (n) | 6 (2–19) | 41 (25–48) | <0.001 |
| P3M (n) | 0 (0–0) | 1 (0–5) | <0.001 |
| FTM (n) | 1 (0–3) | 11 (6–18) | <0.001 |
| REB (n) | 8 (4–19) | 38 (25–62) | <0.001 |
| AS (n) | 3 (0–8) | 22 (10–34) | <0.001 |
| ST (n) | 1 (0–2) | 5 (4–8) | <0.001 |

Data are summarized using the median and, in brackets, the interquartile range for continuous variables, and absolute and percentage relative frequencies for categorical variables. *p*, p-value; min, minutes; PTS, total points made per match (PTS = FTM + 2*P2M + 3*P3M); P2M, number of two-point field-goals made; P3M, number of three-point field-goals made; FTM, number of free-throw points made; REB, number of rebounds; AS, number of assists; ST, number of steals.

Secondly, we compared the performance of the clusters with that of the four functional classes by estimating a regression model that incorporated the minutes played by team players in functional Classes II, III, and IV. The results did not reveal any significant correlations, as indicated by an adjusted r2 of -0.14 and a *p* of 0.59 (Table 4). This indicates that increasing the playing time for players from Classes III and IV does not necessarily improve team performance.

Figs 4 and 5 illustrate the findings discussed above regarding clusters and functional classes, respectively. Fig 4 shows, for the considered teams in the second round, the proportional composition of the two clusters in each team (the height of each bar is proportional to the percentage of minutes played by players in Cluster 1 and 2, using the scale of the vertical axis on the left) and the plus-minus, illustrated by the grey line (according to the vertical axis on the right). Teams are plotted in ascending order according to the percentage of Cluster 2. For example, Team 4 shows 50% of minutes played by players in Cluster 1 and 50% by players in Cluster 2; the plus-minus value for Team 4 equals -100. According to Fig 4, there is an association between the proportional composition of the two clusters in each team and the net performance, as measured by plus-minus.

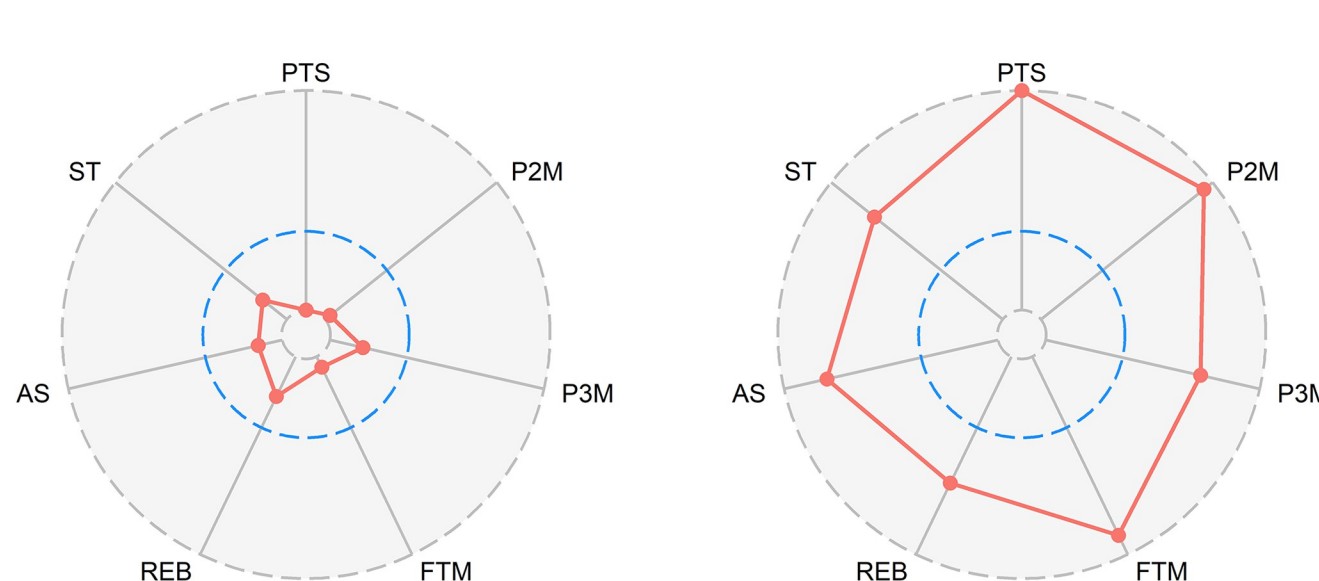

**Fig 2. Profile plots of seven game-related statistics normalized by play time.** The dashed blue line represents the midpoint between the minimum and maximum values. First-round championship data was used for the analysis. PTS, total points made per match (PTS = FTM + 2*P2M + 3*P3M); P2M, number of two-point field-goals made; P3M, number of three-point field-goals made; FTM, number of free-throw points made; REB, number of rebounds; AS, number of assists; ST, number of steals.

Fig 5 shows, for the considered teams in the second round, the proportional composition of the four functional classes in each team (bars, using the scale of the vertical axis on the left) and the plus-minus, illustrated by the grey line (according to the vertical axis on the right). Teams are plotted in ascending order according to the percentage of the fourth functional class. This plot suggests no association between functional classes and the net performance.

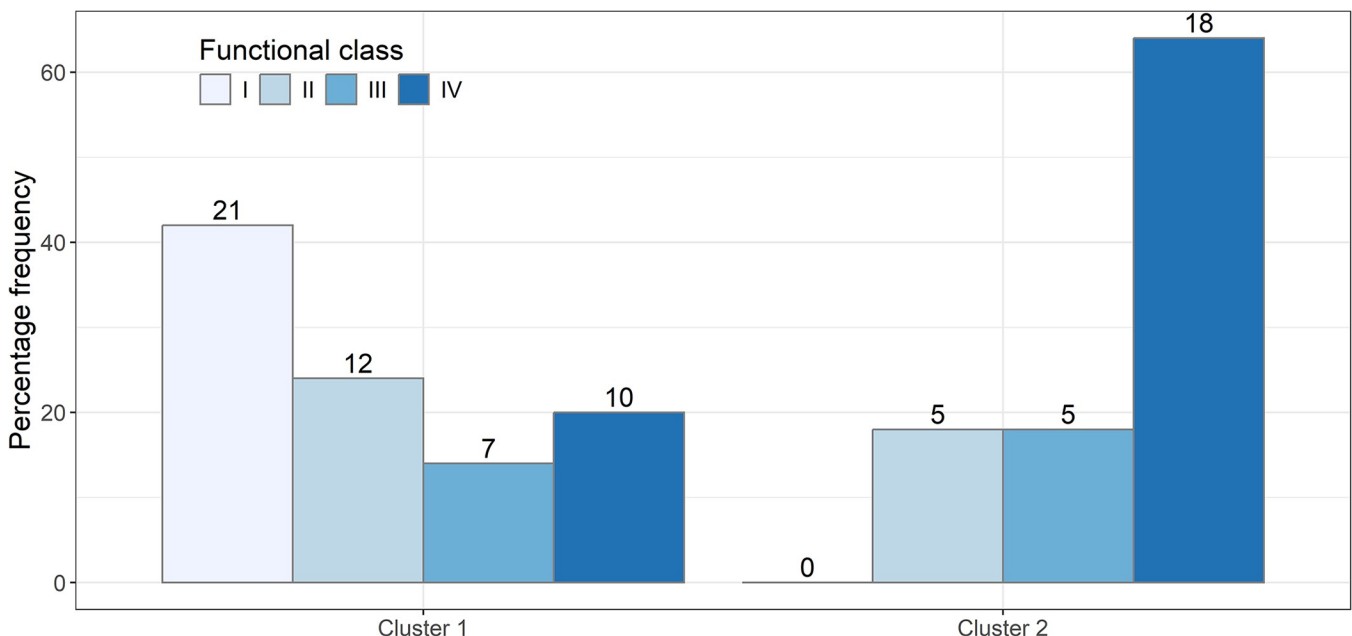

**Fig 3. Distribution of athletes across the four functional classes within the two clusters.**

**Table 4. Linear regression models evaluating the association between team's net performance as the outcome variable, and the minutes played by players of the cluster (Model 1) and in the functional classes (Model 2).**

| Model 1 | | | | |
|---|---|---|---|---|
| Predictor variable | Correlation (95% CI) | $p^{i}$ | Adjusted r2 | $p^{ii}$ |
| Cluster 2 | 0.74 (0.45–0.99) | <0.001 | 0.48 | 0.026 |
| **Model 2** | | | | |
| Predictor variable | Correlation (95% CI) | $p^{i}$ | Adjusted r2 | $p^{ii}$ |
| Functional Class II | 0.28 (-0.84–0.76) | 0.57 | -0.14 | 0.59 |
| Functional Class III | 0.17 (-0.80–0.72) | 0.66 | | |
| Functional Class IV | -0.14 (-0.86–0.64) | 0.68 | | |

95% CI = Percentile bootstrap confidence intervals; $p^{i}$ = $p$ the t test on Pearson's correlation (Model 1); $p^{ii}$ = $p$ the ANOVA F test.

These results highlight that the suggested procedure actually enabled us to select a player exhibiting a stronger association with optimal performance as compared to selections based solely on points, which is typically the primary criterion employed by WB coaches in practice.

## Discussion

This study was designed to provide a practical tool for WB coaches based on statistical techniques to support their tactical decisions in identifying the most influential players to form winning lineups while satisfying the 14-point constraint. In the realm of sporting events,

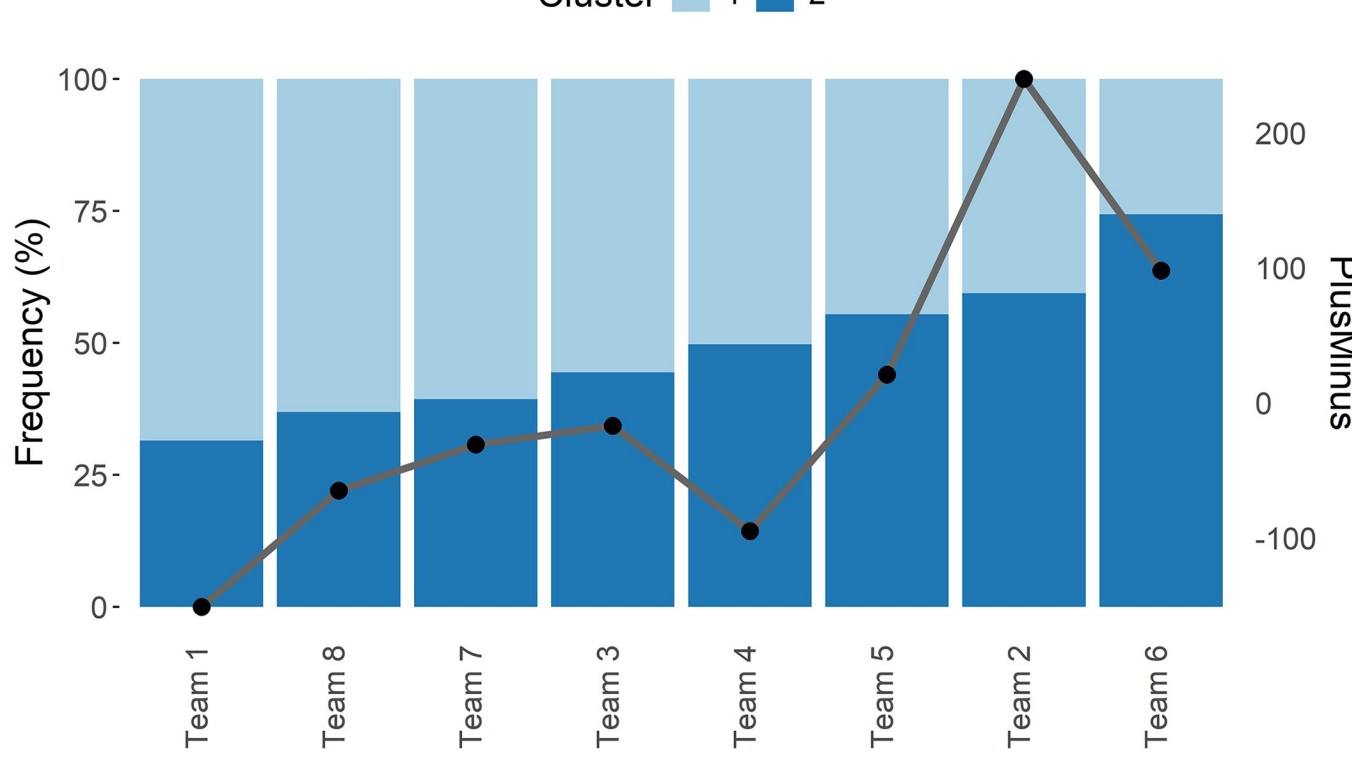

**Fig 4. Performances (plus-minus) of the teams in the second round according to the minutes played by the players belonging to the two clusters identified in the first round.** The bars denote the proportional composition of each team's clusters, arranged in ascending order according to the percentage representation of Cluster 2. The grey line depicts the teams' net performance (PlusMinus), presented in the same order as the bars.

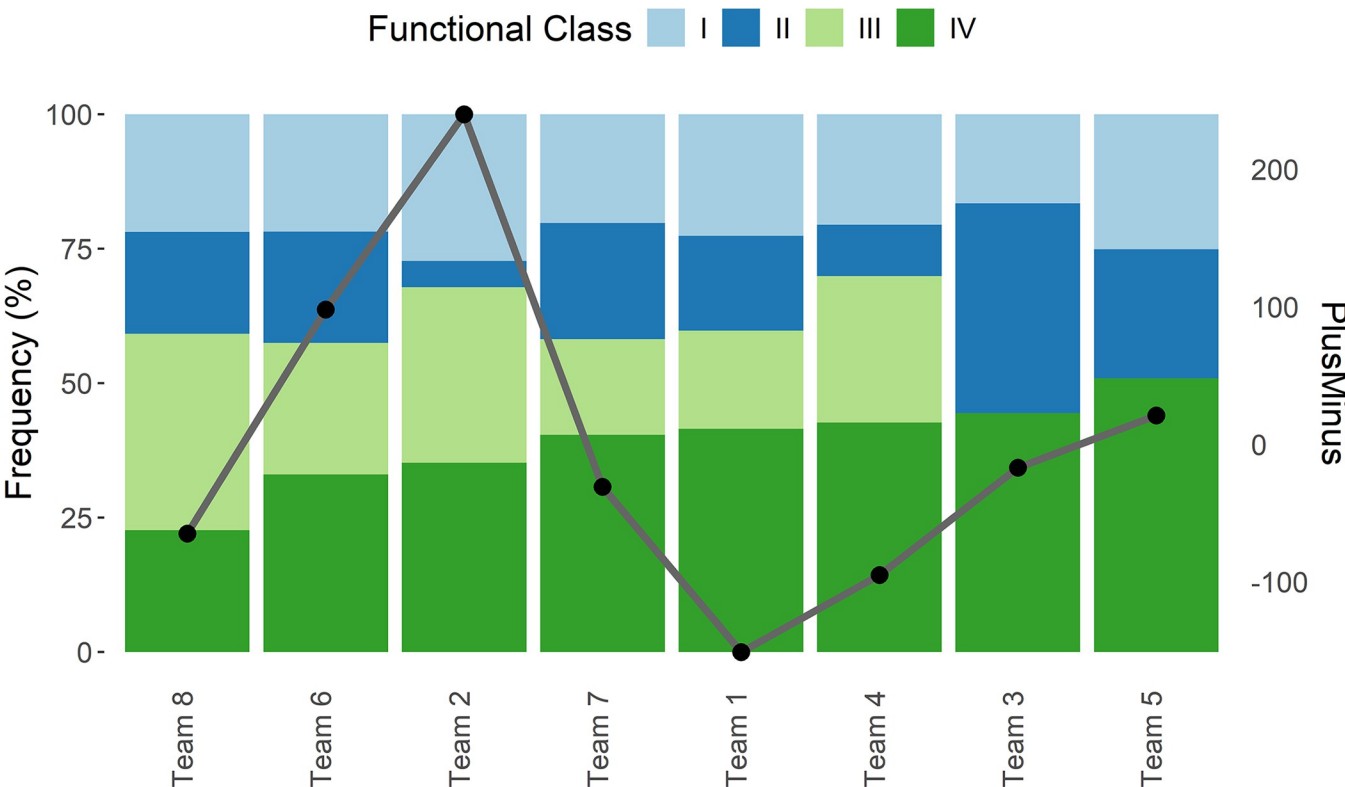

**Fig 5. Performances (plus-minus) of the teams in the second round according to the minutes played by the players in the four functional classes.** The grey line depicts the net performance (PlusMinus) of the teams.

identifying groups with varying performance levels among participants is crucial, as this insight can effectively guide strategic planning. Despite its importance, there is limited literature on the segmentation of participant groups and, more generally, on the application of data science to sport performance in a Paralympic context.

The present preliminary study, motivated by a crucial question from coaches working with wheelchair basketball (WB) players, aimed to provide them with a practical tool based on statistical techniques. This tool supports tactical decision-making by identifying the best-performing players and defining winning lineups within the 14-point constraint. In collaboration with statisticians, this question was translated into testable hypotheses, and statistical methods were then applied to data collected over the course of an entire regular season from the top-tier Italian Wheelchair Basketball Championship. To comprehensively address the research question, the analysis was organized into three primary stages: Clustering, Characterization, and Validation.

In the Clustering stage, game statistics collected during the initial portion of the championship in question were analyzed, leading to the identification of two distinct player profiles: Cluster 1 and Cluster 2. Cluster 1 comprised low-performing players, defined as those who scored lower in terms of total points, rebounds, assists, and steals. Conversely, Cluster 2 was composed of high-performing athletes, i.e. those who registered elevated values in the specified performance metrics. In summary, the performance variables recorded from players during the first round of a top-level national championship enabled the identification of a two-cluster solution, thereby facilitating the classification of athletes into two homogeneous groups based on their in-game performance metrics.

At this point in the analysis, one could hypothesize that the low-performance group would be composed of players with higher functional limitations (i.e., players belonging to Class I or II) and that the high-performance group would be composed of players with lower functional limitations (i.e., players belonging to Class III or IV). In fact, as expected, we observed that players belonging to Class III and IV (i.e., high-point players) performed better in all the considered scores than those belonging to Class I and II (i.e., low-point players). However, an intriguing result of this first part of the analysis was that there is no exact correspondence between clusters and functional classes, in the sense that Cluster 1 (i.e., low-performance players) does not exclusively contain players of Class I and II and correspondingly, Cluster 2 (i.e., high-performance players) does not exclusively contain players of Class III and IV.

Two primary interpretations can be drawn from the above-reported result. First, it confirms the existence of a relationship between functional classification and sport-specific performance in WB, corroborating previous studies [21]. This finding reinforces the notion that performance in WB is influenced by players' residual functional abilities and, consequently, their assigned functional classes. The second interpretation stemming from these findings suggests that an athlete's performance is influenced not only by the type and severity of disability but also by several other factors. These factors include the volume and quality of training, sport experience, predisposing conditions, prior athletic experiences, individual skill sets, talent, motivation, resilience, and emotional management during competitions, among others. Considering that, in a WB game, the combined score of the five players on the court at any time cannot exceed 14 points as per the International Wheelchair Basketball Federation guidelines [6], we hypothesized that coaches must take into account a variety of factors beyond disability when determining game outcomes. These factors may not be solely dictated by the functional class assigned to the players. This observation accentuates the need for coaches and technical staff to have practical tools available to make player selections based on criteria that extend beyond merely the players' assigned functional classes.

In subsequent analyses incorporating game statistics from the latter part of the championship under consideration, the results corroborated the initial findings: players in Cluster 2 outperformed those in Cluster 1. This consistency across both segments of the championship substantiates the initial categorization into high-performing and low-performing players. Such robustness in the data afforded us the confidence to proceed with further analyses and validate the integrity of the two distinct player profiles.

The Validation stage was further subdivided into two steps: the first focused on investigating whether a team's success correlated with coaches allocating more playing time to athletes in Cluster 2, and the second exploring the relationship between extended playing time for players classified in Classes III and IV and the overall team performance. The regression models revealed that teams in which players from Cluster 2 received more playing time achieved better overall performance. However, extending playing time for Classes III and IV athletes did not necessarily correlate with improved team outcomes. These findings suggest that our statistical approach, which identifies high-performing athletes through cluster analysis of game statistics from the first part of a season, could serve as a valuable tool for WB coaches. It could encourage the best possible combinations between functional class and their WB personal sport-specific abilities and it could ultimately maximize the collective performance in a national top-level WB tournament. Specifically, this approach (Fig 1) can help coaches make tactical decisions based on distinct player profiles, thereby optimizing both the effectiveness and efficiency of game management.

The main differences between WB and running basketball which condition the WB players role and possible actions on the court are the use of the wheelchair and the adoption of a functional classification system [34]. In the literature, the functional class is considered a

fundamental aspect when analyzing WB performance. In particular, it has been shown that the functional class is related to the volume of action [6], physical performance [23, 35], and playing role [18, 36]. Along with these aspects, coaches must also consider the 14 point limit [6] of the functional class total. When choosing the players that will make up the final squad for a given competition coaches have to try to create the best possible combinations between players and their functional classes to maximize team performance in that specific game [18]. Of course, there are multiple possibilities in the composition of the WB lineups, conditioned by the availability of players from different functional classes in the team. Today, there is not enough evidence regarding the performance of a team lineup, analyzed throughout game statistics, to determine which are the most frequent with regards to the type of game, competitive level, and phase of the competition. The present study represents one of the few studies [34] investigating strategies to configure team lineups and promoting the employment of game statistics.

This study has some limitations that are important to underline. First, we considered only one national championship. In future research, it would be interesting to assess whether these results are confirmed when taking into consideration more championships (i.e., more consecutive competitive seasons and data extracted from championships in other countries). Additionally, our analysis was limited to box scores game statistics; a more in-depth exploration of analytics could yield further insights. In future research, it would be interesting to collect play-by-play data for WB players as well. So, new analysis approaches would be opened, such as the possibility of normalizing statistics with the team pace when each player was on the court instead of minutes played.

## Conclusions

In conclusion, the statistical approach proposed in this study enabled us to select players based on performance compared to selections based solely on points, which is the usual criterion employed in practice. These results appear to present important opportunities for further analysis. It has the potential to increase success in WB matches and could be a key factor in improving game outcomes if employed.

The present preliminary study, motivated by a crucial question from coaches working with WB players, aimed to provide them with a practical tool based on statistical techniques. This tool supports tactical decision-making by identifying the best-performing players and defining winning lineups within the 14-point constraint. In the future, it will be important to gather more comprehensive data from WB games, including advanced analytics, play-by-play feedback, and so on, to gain a multifaceted understanding of this Paralympic sport. However, it is important to bear in mind that game statistics are data that, when collected in large quantities and appropriately analyzed, should be transformed into nothing more than useful information to support technical experts in their decisions. In fact, the authors of this study strongly believe that while data from game statistics are indeed a useful tool, relying solely on these statistics to evaluate performance would be both reductive and misleading. Such an approach would reduce the game to numbers that cannot fully capture its complexities. In other words, these statistical tools serve as a support for decision-making but cannot replace the expertise and direct experience of WB coaches.

In conclusion, to enhance WB performance, collaboration between WB experts and data scientists is crucial. In this partnership, WB experts should identify a problem and pose relevant research questions, while data scientists apply their statistical expertise to carry out the analyses. This collaborative approach between professionals would be significant for the advancement of a growing Paralympic sport like WB. It can culminate in coaches routinely using information and knowledge gleaned from data science to inform their decision-making.

## Supporting information

**S1 File. Supplemental information for this article.**
(CSV)

## Acknowledgments

The authors would like to thank the Federazione Italiana Pallacanestro in Carrozzina (FIPIC) for their kind support.

## Author Contributions

**Conceptualization:** Paola Zuccolotto, Chiara Milanese.

**Data curation:** Valentina Cavedon, Paola Zuccolotto, Marco Sandri, Maricay Manisera.

**Formal analysis:** Valentina Cavedon, Paola Zuccolotto, Marco Sandri, Maricay Manisera.

**Investigation:** Valentina Cavedon.

**Methodology:** Valentina Cavedon, Paola Zuccolotto, Marco Sandri, Maricay Manisera, Marco Bernardi, Ilaria Peluso, Chiara Milanese.

**Supervision:** Marco Bernardi, Ilaria Peluso, Chiara Milanese.

**Writing – original draft:** Valentina Cavedon, Paola Zuccolotto, Marco Sandri, Chiara Milanese.

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
