## [Decision Letter · Decision Letter 0]

4 Jan 2024

PONE-D-23-40602Coaching strategies in wheelchair basketball: a statistical approach for player selection on the courtPLOS ONE

Dear Dr. Cavedon,

Thank you for submitting your manuscript to PLOS ONE. After careful consideration, we feel that it has merit but does not fully meet PLOS ONE’s publication criteria as it currently stands. Therefore, we invite you to submit a revised version of the manuscript that addresses the points raised during the review process.

We look forward to receiving your revised manuscript.

Kind regards,

Gianpiero Greco

Academic Editor

PLOS ONE

Journal Requirements:

Reviewers' comments:

Reviewer's Responses to Questions

**Comments to the Author**

1. Is the manuscript technically sound, and do the data support the conclusions?

Reviewer #1: No

Reviewer #2: Partly

2. Has the statistical analysis been performed appropriately and rigorously? 

Reviewer #1: No

Reviewer #2: Yes

3. Have the authors made all data underlying the findings in their manuscript fully available?

Reviewer #1: No

Reviewer #2: Yes

4. Is the manuscript presented in an intelligible fashion and written in standard English?

Reviewer #1: Yes

Reviewer #2: No

5. Review Comments to the Author

Reviewer #1: As the title of the article and the introduction suggest, this study is about what research variables could be used to select players for WB matches.

What the main research question is remains unclear to me. Maybe this one “By what criteria can I select which players to put on the field during a WB match?”. However, this question can be solved in a practical rather than a scientific way. Because its validity had to be verified by scientific methods. If it is intended to be done, it must be named. It is not specified what the aim of the study is, it is only stated “We develop a method that enables to recognize the best-performing players...” But what is the main point of doing this?

The authors did not fulfilled how to describe the research sample, some information moved to the results section. Not sure how many games each team played? If there are 56 matches and 8 teams, then they played 7 matches each. Why it is said that after 4? It is not clear how it was possible to play 7 or 4 games in 14 days?

The literature review is done well, but what is the scientific and not the practical justification for this work?

Statistical values of positions should be expressed as means and SD. This would allow us to understand the non-absolute values of each sample and could be compared with other data according to existing scientific standards

The results section should clearly state what you got in response to the main research question. This information is not available.

The statement about the conclusions sounds like this “In conclusion, the statistical approach proposed in this study appears to present significant opportunities for further analysis”. However, what specific conclusion is made that would allow us to answer what valid scientific method allows us to talk about player selection during matches?

The conclusions section must be separate.

There is a lot of talk about practical use. A separate section of the article should be assigned to it.

Reviewer #2: Dear Authors,

I would like to express my gratitude for the opportunity to review this manuscript.

The manuscript at this stage requires improvements. Below are suggestions with line indications:

21 – Abbreviations in the manuscript for the first time should be in full.

28 & 30 – Please correct the “R-squared” and “p-value” format.

34-35 – Please correct the keywords format, considering the journal template and instructions for authors.

39 – Please correct the citations format, not only in this line, but in all manuscript. Please consider the journal template and instructions for authors.

38-55 - Please consider shorter paragraphs (8-12 lines) to increase readability. This suggestion applies to all manuscript.

78-88 – The end of the introduction section should clearly state the aim of the study.

117 – Please revise the text format.

172-173 – Please remove one line.

164, 190 – Tables titles are too long, please consider shorter titles.

194 – Please consider a table footnote.

196-203 – It is not understandable the content in these lines. Please clarify.

225 – Please consider the journal template and instructions for authors regarding all table's format.

235, 236 – Please consider each figure introduction previously to its appearance in the manuscript, and also a short analysis after each figure.

260 – This content is part of the methodology, not the discussion section. Please consider reformulation.

249 – All the discussion section should be reformulated. Starting by indicating the aim of the study, and afterward presenting the main findings of the study, comparing these afterward with the literature. References are suggested to be presented in this section. The last paragraph should highlight the study limitations and suggestions for future research.

339 – Content is missing before references, please check the journal template.

340 – All references should be carefully revised, they are not according to the journal template and instructions for authors.

Please carefully revise the English.

6. PLOS authors have the option to publish the peer review history of their article (what does this mean?). If published, this will include your full peer review and any attached files.

Reviewer #1: No

Reviewer #2: No

---

## [Author Response · Author response to Decision Letter 0]

11 Mar 2024

Valentina Cavedon

Department of Neurosciences, Biomedicine and Movement Sciences

University of Verona, Italy

Verona, 11/03/2024

Revision: Manuscript “Coaching Strategies in Wheelchair Basketball: A Statistical Approach for Player Selection on the Court” (title in the revised manuscript: Optimizing Wheelchair Basketball Lineups: A Statistical Approach to Coaching Strategies”) by Valentina Cavedon, Paola Zuccolotto, Marco Sandri, Marica Manisera, Marco Bernardi, Ilaria Peluso, Chiara Milanese.

Dear Professor Gianpietro Greco 

Gianpiero Greco

Academic Editor

PLOS ONE

I am submitting on behalf of all authors the revised version of the quoted manuscript for your and the reviewers’ consideration.

The authors would like to thank you and the reviewers for their valuable comments and their time. The constructive criticism from the reviewers was appreciated and the manuscript accordingly modified. Modifications and insertions were made to the text to improve the presentation of the work, clarifying all the points that were requested. All of the points made by the reviewers were carefully taken into consideration as we hope you will see from the modifications made and my notes on the revision.

Three files are submitted: a first one, labelled “Revised manuscript with tracked changes” is the original main document with tracking of changes; a second one, labelled “Manuscript” is the final resubmitted version of the manuscript. A third file, labelled “Rebuttal letter” is the point-by-point rebuttal letter to the Reviewers.

In the Acknowledgments section, we included the following sentence “Research Project PRIN 2022, granted by European Union – Next Generation EU, “Statistical Models and AlgoRiThms in sports (SMARTsports). Applications in professional and amateur contexts, with able-bodied and disabled athletes”, project nr. 2022R74PLE, CUP: D53D23005950006.”

We consider the revised and improved manuscript to be complete and we hope that the manuscript is now suitable for publication in Plos One.

Thank you for your kind attention.

Sincerely Yours,

Valentina Cavedon

Correspondence to:

Valentina Cavedon, Department of Neurosciences, Biomedicine and Movement Sciences, University of Verona, 37134 Verona, Italy.

Strada Le Grazie, 8.

Tel. +39-0458425173 Fax: +39-0458425131

E-mail: valentina.cavedon@univr.it.

Reviewer #1: 

As the title of the article and the introduction suggest, this study is about what research variables could be used to select players for WB matches.

What the main research question is remains unclear to me. Maybe this one “By what criteria can I select which players to put on the field during a WB match?”. However, this question can be solved in a practical rather than a scientific way. Because its validity had to be verified by scientific methods. If it is intended to be done, it must be named. It is not specified what the aim of the study is, it is only stated “We develop a method that enables to recognize the best-performing players...” But what is the main point of doing this?

The author would like to thank the Reviewer for his/her time to review the manuscript and for his/her valuable suggestions. We appreciate the reviewer's feedback, which prompted us to reconsider and refine our paper's objective. Upon reflection, the original research question, 'By what criteria should players be selected for participation in a WB match?' was open to interpretation. This feedback has led us to rephrase our objective to improve clarity and facilitate better understanding.

The goal of the study is not to suggest which variables can be used to select the best players to be employed during matches, nor is it to propose a predictive model of the team’s final performance. Instead, the objective is to propose a statistical procedure, an algorithm, consisting of three steps (Clustering, Characterization, Validation) that can help identify the most suitable players for field placement, which may not necessarily be those with the highest IWBF. 

The proposed cluster analysis, applied to all players in the championship at the end of the first half of the season, allows for the classification of players into two groups based on their on-field performance, measured by traditional game variables. The two groups separate the more and the less skilled players, providing the coach with concise information (the group membership of each player, including players from opposing teams) to form the best lineup while still satisfying the 14-point constraint.

An interesting aspect of the classification obtained is that the group of better players includes some individuals with low IWBF scores, while the group of worse players contains some with high IWBF scores.

The proposed cluster analysis aims to create groups of players that are (i) homogeneous, meaning they consist of similar players with regard to the considered game variables (minimal within-cluster variability), and simultaneously (ii) well separated, whereby players belonging to different groups are dissimilar with respect to the considered game variables (maximal between-cluster variability). This objective is attained by using all traditional game variables measured across all championship players (and not only the players of one single team), ensuring an adequate dataset for analysis. While this may initially appear challenging, it is not, as the data used in the analysis are typically readily recoverable.

Validation is conducted by studying the relationship between the composition of the individual team and the team's final performance, as measured by total points. This analysis, using linear regression models and Figures 4 and 5, shows that teams using more minutes or players from the cluster of better players achieve better performances.

This validation allows us to state that the proposed procedure for selecting players for the team works better than the choice based on IWBF, which is typically used by coaches since it is implicit in the definition of IWBF and the existence of the 14-point constraint.

We modified the manuscript as follows.

First, we modified the title as follows:

Optimizing Wheelchair Basketball Lineups: A Statistical Approach to Coaching Strategies

Lines 18-19

We replaced the sentence “this study was designed…. a match?” with the following:

“This study was designed to support the tactical decisions of wheelchair basketball (WB) coaches in identifying the best players to form winning lineups.” 

Lines 82-83

We replaced the sentence “… to support their tactical decisions in order to answer …. match?” with the following:

“…this study was designed to provide a practical tool for WB coaches based on statistical techniques to support their tactical decisions in identifying the most influential players to form winning lineups while satisfying the 14-point constraint.”

Line 123

To emphasize the advantage of using cluster analysis, we added the following sentence:

“Cluster analysis enables the selection of top players based on multiple variables simultaneously, a task that exceeds the capabilities of the human brain due to its inherently multivariate nature.” 

Line 236

We clarified the obtained result by adding the following sentence:

“These results highlight that the suggested procedure enabled us to obtain a player selection exhibiting a stronger association with optimal performance compared to selections based solely on Points, which is typically the primary criterion employed by WB coaches in practice.”

Lines 253-255

We replaced the sentence “The present preliminary study … WB match?” with the following:

“The present preliminary study was driven by a crucial question from coaches working with WB players to provide them with a practical tool based on statistical techniques to support their tactical decisions in identifying the best-performing players and to define winning lineups within the 14-point constraint.” 

The authors did not fulfilled how to describe the research sample, some information moved to the results section. Not sure how many games each team played? If there are 56 matches and 8 teams, then they played 7 matches each. Why it is said that after 4? It is not clear how it was possible to play 7 or 4 games in 14 days?

More information and clarification have been provided in the Participants section. Moreover, in the characteristics of the players have been moved from the Results section to the Participants section. 

The literature review is done well, but what is the scientific and not the practical justification for this work?

The question concerning the scientific and practical justification of our work is interesting and relevant. 

In the initial planning stages of our investigation, the foremost objective we established was to devise an approach for coaches that is not only simple and practical for everyday use but also firmly rooted in the principles of statistical analysis, thereby ensuring a solid scientific foundation. This aim guided our research methodology and influenced the development of the proposed tool.

Nonetheless, from a scientific perspective, our study reveals some interesting findings: 

- it is possible to identify a subgroup of players whose performance is significantly superior to others 

- the teams that make the most use of these players on the field tend to have higher total scores at the end of the season (this association does not hold when analyzing the association between the total scores and the utilization of IWBF Class IV players)

- this subgroup of players only partially overlaps with the Functional Class IV of the International Wheelchair Basketball Federation (IWBF); indeed, within the subgroup, we find that 36% of the players are from Class III and II.

Statistical values of positions should be expressed as means and SD. This would allow us to understand the non-absolute values of each sample and could be compared with other data according to existing scientific standards.

After careful consideration, we respectfully acknowledge that the reviewer's point might not have been entirely clear to us. 

Firstly, the term 'position' as used by the reviewer is ambiguous to us. Is he/she referring to the mean as a measure of central tendency? Are we being suggested to alter our method of summarizing the continuous variables in Table 1, possibly using the mean and standard deviation instead of the median and IQR? If so, we would like to respectfully point out that these variables are predominantly count variables (counting points, shots, seconds/minutes elapsed). As per statistical principles, the median and IQR are more appropriate measures of central tendency and dispersion for such data than the mean and SD.

The second point that is unclear to us is how the reviewer considers the mean and SD to be 'non-absolute values'. The mean and SD have the same units of measurement as the variable they describe, and thus are technically 'absolute values' (examples of 'non-absolute values' could be ratios or differences). Or perhaps is the reviewer advising us to use mean and SD because they are often used in the literature (and thus could be considered 'scientific standards'), which would allow the results of our study to be comparable with those reported in other scientific works?

The results section should clearly state what you got in response to the main research question. This information is not available.

The statement about the conclusions sounds like this “In conclusion, the statistical approach proposed in this study appears to present significant opportunities for further analysis”. However, what specific conclusion is made that would allow us to answer what valid scientific method allows us to talk about player selection during matches?

Line 322

We clarified the obtained result by replacing the sentence:

“In conclusion, … further analysis” with the following:

“In conclusion, the statistical approach proposed in this study enabled us to obtain a player selection with better performance compared to selections based solely on Points, which is the usual criterion employed in practice.”

The conclusions section must be separate.

There is a lot of talk about practical use. A separate section of the article should be assigned to it.

The conclusion section has been separated from the Discussion section, as suggested by the Reviewer.

Reviewer #2: 

Dear Authors,

I would like to express my gratitude for the opportunity to review this manuscript.

The manuscript at this stage requires improvements. 

Below are suggestions with line indications:

The author would like to thank the Reviewer for his/her time to review the manuscript and for his/her valuable suggestions.

21 – Abbreviations in the manuscript for the first time should be in full.

Modification has been made. In the revised manuscript, “WB” reads “Wheelchair Basketball (WB)”. 

28 & 30 – Please correct the “R-squared” and “p-value” format.

We are sorry, but “R-squared” and “p-value” read correctly in our PDF version. What is the error in the format you are referring to?

34-35 – Please correct the keywords format, considering the journal template and instructions for authors.

The keywords have been deleted from the Manuscript according to the instructions for authors. 

39 – Please correct the citations format, not only in this line, but in all manuscript. Please consider the journal template and instructions for authors.

The references have been formatted according to the “PlosONE” style (i.e., Vancouver style) and downloaded for Zootero according to the instructions for authors. 

38-55 - Please consider shorter paragraphs (8-12 lines) to increase readability. This suggestion applies to all manuscript.

In the revised Manuscript, the paragraphs are shorter (i.e., 8-12 lines). 

78-88 – The end of the introduction section should clearly state the aim of the study.

The aim has been more clearly stated at the end of the Introduction section. 

117 – Please revise the text format.

The sentence has been reformulated for clarity. It now reads, " The analysis performed in our study consisted of three main steps: the Clustering step, the Characterization step, and the Validation step.”

172-173 – Please remove one line.

One line has been removed.

164, 190 – Tables titles are too long, please consider shorter titles.

The Titles of the Tables have been shortened.

194 – Please consider a table footnote.

A Table footnote has been added. 

196-203 – It is not understandable the content in these lines. Please clarify. 

The Figure Captions in the revised Manuscript have been modified according to the instructions for authors (https://journals.plos.org/plosone/s/file?id=wjVg/PLOSOne_formatting_sample_main_body.pdf).

225 – Please consider the journal template and instructions for authors regarding all table's format.

Tables have been modified according the journal template

(https://journals.plos.org/plosone/s/file?id=wjVg/PLOSOne_formatting_sample_main_body.pdf).

235, 236 – Please consider each figure introduction previously to its appearance in the manuscript, and also a short analysis after each figure.

We improve the figures’ description.

Line 177 (after “Figure 1”):

We added: “The left radial plot illustrates that all variable values for players within Cluster 1 are lower compared to those in Cluster 2, as depicted in the right-hand radial plot. Indeed, ….”

Line 186 (after “Figure 2”):

We added: “The two clusters show different distributions.” 

Line 236 (after “respectively”):

We added: “Figure 4 shows, for the considered teams in the second round, the proportional composition of the two clusters in each team (the height of each bar is proportional to the percentage of minutes played by players in Cluster 1 and 2, using the scale of the vertical axis on the left) and the plus-minus, illustrated by the grey line (according to the vertical axis on the right). Teams are plotted in ascending order according to the percentage of Cluster 2. For example, Team 4 shows 50% of minutes played by players in Cluster 1 and 50% by players in Cluster 2; the plus-minus value for Team 4 equals -100. According to Figure 4, there is an association between the proportional composition of the two 

---

## [Decision Letter · Decision Letter 1]

24 Mar 2024

PONE-D-23-40602R1Optimizing Wheelchair Basketball Lineups: A Statistical Approach to Coaching StrategiesPLOS ONE

Dear Dr. Cavedon,

Thank you for submitting your manuscript to PLOS ONE. After careful consideration, we feel that it has merit but does not fully meet PLOS ONE’s publication criteria as it currently stands. Therefore, we invite you to submit a revised version of the manuscript that addresses the points raised during the review process.

We look forward to receiving your revised manuscript.

Kind regards,

Gianpiero Greco

Academic Editor

PLOS ONE

Journal Requirements:

Reviewers' comments:

Reviewer's Responses to Questions

**Comments to the Author**

1. If the authors have adequately addressed your comments raised in a previous round of review and you feel that this manuscript is now acceptable for publication, you may indicate that here to bypass the “Comments to the Author” section, enter your conflict of interest statement in the “Confidential to Editor” section, and submit your "Accept" recommendation.

Reviewer #1: All comments have been addressed

Reviewer #2: All comments have been addressed

2. Is the manuscript technically sound, and do the data support the conclusions?

Reviewer #1: Yes

Reviewer #2: Partly

3. Has the statistical analysis been performed appropriately and rigorously? 

Reviewer #1: Yes

Reviewer #2: Yes

4. Have the authors made all data underlying the findings in their manuscript fully available?

Reviewer #1: Yes

Reviewer #2: Yes

5. Is the manuscript presented in an intelligible fashion and written in standard English?

Reviewer #1: Yes

Reviewer #2: No

6. Review Comments to the Author

Reviewer #1: (No Response)

Reviewer #2: Dear Authors,

Thank you for considering my suggestions and incorporating them into the manuscript, which is globally improved, congratulations.

The manuscript at this stage still requires improvements. Below are suggestions with line indication:

32 – Please consider “p” in italic in all manuscript (including tables). “R-squared” can also be considered r2. Please revise both cases.

37-38 – The keywords should not be deleted but written according to the journal instructions for authors.

99 – Please clearly state the inclusion and exclusion criteria.

114-120 – Please consider a table to provide this information to the readers.

Between “sample” and “statistical analysis”, please consider subtopics such as “study design” and “procedures” associated with the presentation of relevant information to the readers.

131 – Please revise the text of “statistical analysis”. Some paragraphs are too short and line 164 should be deleted.

202-203 – Please revise.

208 – It is suggested to present Figure 2 after the respective introduction. The same should be considered for table 2 (L 206), Figure 4 (L 287). Please consider reformulating this section (“results”).

225 – Please format the tables considering the journal instructions for authors (text type, size, titles, and footnotes), and aiming to standardize the text format (namely text in the same line).

308 – Despite the improvement of the discussion section, particularly with the inclusion of new references, some paragraphs are too long, which difficult to read and analysis. Please revise this section.

409 - Please consider paragraphs aiming for more clear/direct take-home messages in the conclusions section. Please also consider some practical application text.

441 – Please double-check the references format details. For example, the ref 5 title is in lowercase, contrary to others; Ref 15 journal format should be corrected. Please revise all refs in detail.

The English improved, congratulations. Nevertheless, please carefully revise the new version of the manuscript.

7. PLOS authors have the option to publish the peer review history of their article (what does this mean?). If published, this will include your full peer review and any attached files.

Reviewer #1: No

Reviewer #2: No

---

## [Author Response · Author response to Decision Letter 1]

5 Apr 2024

Reviewer #1: (No Response)

Reviewer #2: 

Dear Authors, thank you for considering my suggestions and incorporating them into the manuscript, which is globally improved, congratulations. The manuscript at this stage still requires improvements. Below are suggestions with line indication:

The authors would like to thank the Reviewer for their time and valuable suggestions. 

32 – Please consider “p” in italic in all manuscript (including tables). “R-squared” can also be considered r2. Please revise both cases.

In the revised manuscript “p-value” is “p” and “R-squared” is “r2”.

37-38 – The keywords should not be deleted but written according to the journal instructions for authors.

We apologize, but we did not understand this request from the reviewer. 

We have looked for guidelines regarding the keywords in the journal's instructions but found nothing. We also reviewed many recent papers published by PLOS One, but none of them report keywords. Even the template provided by the journal does not mention keywords. 

Therefore, we are asking for instructions on how to include them in the text of our paper. Thank you.

99 – Please clearly state the inclusion and exclusion criteria.

The inclusion criteria have been detailed in the Materials and Methods section.

114-120 – Please consider a table to provide this information to the readers.

We have added a table containing this information.

Between “sample” and “statistical analysis”, please consider subtopics such as “study design” and “procedures” associated with the presentation of relevant information to the readers.

In the Materials and Methods section, subtopics such as “study design” and “procedures” have been included.

131 – Please revise the text of “statistical analysis”. Some paragraphs are too short and line 164 should be deleted.

Line 164 has been deleted. The text of "Statistical Methods" has been improved, some paragraphs that were too short have been reformulated, and small changes have been made to enhance clarity.

202-203 – Please revise.

Done. The two blank lines have been removed. 

208 – It is suggested to present Figure 2 after the respective introduction. The same should be considered for table 2 (L 206), Figure 4 (L 287). Please consider reformulating this section (“results”).

The Results section has been reformulated, reporting Figure 2, Table 2 (now Table 3) and Figure 4 after the respective introduction paragraphs.

225 – Please format the tables considering the journal instructions for authors (text type, size, titles, and footnotes), and aiming to standardize the text format (namely text in the same line).

The Tables have been formatted according to the instructions for authors reported on the PlosOne site. 

308 – Despite the improvement of the discussion section, particularly with the inclusion of new references, some paragraphs are too long, which difficult to read and analysis. Please revise this section.

In the revised manuscript, the Discussion and Conclusion sections include shorter paragraphs to favour readability.

409 - Please consider paragraphs aiming for more clear/direct take-home messages in the conclusions section. Please also consider some practical application text.

In the revised manuscript, the Conclusions section includes shorter paragraphs to favour readability. Some practical applications have been added to the text.

441 – Please double-check the references format details. For example, the ref 5 title is in lowercase, contrary to others; Ref 15 journal format should be corrected. Please revise all refs in detail.

The references have been double-checked, and all references have been revised.

The English improved, congratulations. Nevertheless, please carefully revise the new version of the manuscript.

The revised manuscript has been edited by a native English speaker.

---

## [Editor Report · Decision Letter 2]

9 Apr 2024

Optimizing Wheelchair Basketball Lineups: A Statistical Approach to Coaching Strategies

PONE-D-23-40602R2

Dear Dr. Cavedon,

We’re pleased to inform you that your manuscript has been judged scientifically suitable for publication and will be formally accepted for publication once it meets all outstanding technical requirements.

Kind regards,

Gianpiero Greco

Academic Editor

PLOS ONE